# SIRT3 Deficiency Enhances Ferroptosis and Promotes Cardiac Fibrosis via p53 Acetylation

**DOI:** 10.3390/cells12101428

**Published:** 2023-05-19

**Authors:** Han Su, Aubrey C. Cantrell, Jian-Xiong Chen, Wei Gu, Heng Zeng

**Affiliations:** 1Department of Pharmacology and Toxicology, University of Mississippi Medical Center, Jackson, MS 39216, USAjchen3@umc.edu (J.-X.C.); 2Institute for Cancer Genetics, Columbia University, 1130 Nicholas Avenue, New York, NY 10032, USA; wg8@cumc.columbia.edu

**Keywords:** cardiac fibrosis, myofibroblasts, SIRT3, ferroptosis, p53 acetylation

## Abstract

Cardiac fibrosis plays an essential role in the development of diastolic dysfunction and contributes to heart failure with preserved ejection fraction (HFpEF). Our previous studies suggested Sirtuin 3 (SIRT3) as a potential target for cardiac fibrosis and heart failure. In the present study, we explored the role of SIRT3 in cardiac ferroptosis and its contribution to cardiac fibrosis. Our data showed that knockout of SIRT3 resulted in a significant increase in ferroptosis, with increased levels of 4-hydroxynonenal (4-HNE) and downregulation of glutathione peroxidase 4 (GPX-4) in the mouse hearts. Overexpression of SIRT3 significantly blunted ferroptosis in response to erastin, a known ferroptosis inducer, in H9c2 myofibroblasts. Knockout of SIRT3 resulted in a significant increase in p53 acetylation. Inhibition of p53 acetylation by C646 significantly alleviated ferroptosis in H9c2 myofibroblasts. To further explore the involvement of p53 acetylation in SIRT3-mediated ferroptosis, we crossed acetylated p53 mutant (p53^4KR^) mice, which cannot activate ferroptosis, with SIRT3KO mice. SIRT3KO/p53^4KR^ mice exhibited a significant reduction in ferroptosis and less cardiac fibrosis compared to SIRT3KO mice. Furthermore, cardiomyocyte-specific knockout of SIRT3 (SIRT3-cKO) in mice resulted in a significant increase in ferroptosis and cardiac fibrosis. Treatment of SIRT3-cKO mice with the ferroptosis inhibitor ferrostatin-1 (Fer-1) led to a significant reduction in ferroptosis and cardiac fibrosis. We concluded that SIRT3-mediated cardiac fibrosis was partly through a mechanism involving p53 acetylation-induced ferroptosis in myofibroblasts.

## 1. Introduction

Heart failure with preserved ejection fraction (HFpEF) accounts for more than half of all heart failure cases at present, causing high morbidity and mortality worldwide [1]. Accumulating evidence indicates that most elderly individuals with HFpEF suffer from hypertension, obesity and diabetes [2]. These diseases caused by aging are major drivers of the pathophysiology of HFpEF [2]. Diastolic dysfunction plays an essential role during the development of HFpEF, and cardiac fibrosis is defined as the essential structural alteration contributing to diastolic dysfunction [3]. It is urgent to find therapeutic targets for cardiac fibrosis in aging-associated diseases.

Most studies on fibrosis have been explored from the standpoint of cardiomyocytes or endothelial cells [4,5], with few focusing on fibrotic cells themselves. Myofibroblasts act as a central character in fibrosis, as the secretion of the ECM (extra cellular matrix) represents its main mechanism in most studies [6]. Meanwhile, recent research demonstrated that myofibroblast cell death might also play an important function [7]. Other than apoptosis, autophagy and pyroptosis, ferroptosis was first described by Dixon and is extensively investigated in cardiovascular diseases [8,9,10]. Lipid peroxidation (LPO), as the key characteristic of ferroptosis, produces 4-hydroxynonenal (4-HNE), which contributes to tissue injuries, and transforming growth factor-β1 (TGF-β1), which plays a regulatory role in fibrosis [11,12].

SIRT3, as a key regulator for deacetylation in mitochondria, can influence multitudes of molecular processes, such as metabolic homeostasis, oxidative stress and cell death [13,14]. Interestingly, SIRT3 is called a longevity gene due to its negative correlation with hypertension, obesity and diabetes [15,16], and insufficiency of SIRT3 acts as a contributor to cardiac fibrosis [17,18], which strongly implicates SIRT3 as a crucial mediator between aging-associated diseases and cardiac fibrosis. Moreover, SIRT3 has been shown to regulate ferroptosis via the AMPK/mTOR and SOD2 signaling pathways [19,20]. However, the interactions between SIRT3 and ferroptosis in myofibroblasts need further exploration. P53 is thought of as a cell death regulator mainly via cell-cycle arrest/senescence, such as apoptosis and autophagy [21,22]. In recent years, p53 was proposed by Gu. et al. as a regulator for ferroptosis [23], and its acetylation plays a major role in ferroptosis [23,24]. Acetylation-deficient p53^4KR^ mutant mice were protected from ferroptosis. SIRT3 acts as the key controller of deacetylation, and its absence was found to induce hyperacetylation of p53 [25], which implies a possible association between SIRT3, p53 acetylation and ferroptosis. Whether acetylated p53 acts as a critical mediator between SIRT3 and ferroptosis in myofibroblasts is largely unknown. In the present study, we tested our hypothesis that a loss of SIRT3-induced p53 acetylation, which activates the ferroptosis pathway of myofibroblasts, contributes to cardiac fibrosis and cardiac hypertrophy in mice.

## 2. Materials and Methods

This experiment conformed to the National Institutes of Health (NIH, Bethesda, MD, USA) Guide for the Care and Use of Laboratory Animals (NIH Pub. No. 85-23, Revised 1996). All steps conformed to the Institute for Laboratory Animal Research Guide for the Care and Use of Laboratory Animals. This study was approved by the Animal Care and Use Committee of the University of Mississippi Medical Center (Protocol ID: 1564 and 1189).

### 2.1. Experimental Animal Model and Treatment

Wild-type (WT) SIRT3 control mice, SIRT3 knockout (SIRT3KO) mice and Myh6-Cre transgenic mice were obtained from Jackson Laboratory (Jackson Laboratory, Bar Harbor, ME, USA). P53^4KR^ mice were given by Wei Gu’s lab. We generated SIRT3-cKO mice (Myh6-SIRT3^flox/flox^) by crossing SIRT3^flox/flox^ mice with Myh6-Cre transgenic mice. We generated SIRT3KO/p53^4KR^ mice by crossing SIRT3KO mice with p53^4KR^ mice. These mice were fed normal chow and water. Male mice at age 5–8 months were chosen to perform experiments. We chose male mice instead of female mice since female SIRT3KO mice have less cardiac fibrosis and cardiac dysfunction. SIRT3-cKO mice were treated with saline or ferrostatin-1 (Fer-1, 2 μg/gm, Sigma Aldrich, Saint Louis, MO, USA) every day for 14 days via intraperitoneal injection [9] Experimental mice were euthanized by isoflurane overdose and cervical dislocation under anesthesia.

### 2.2. Cell Culture and Treatment

The H9c2 cell lines were obtained from American Type Culture Collection (ATCC, Manassas, VA, USA). Standard DMEM-basic was used for H9c2 cell culture. These cells were maintained at 37 °C and 5% CO_2_ [26]. H9c2 cells were treated with/without ferroptosis inducer erastin (5 μM, Sigma Aldrich, Saint Louis, MO, USA) and C646 (1–3 μM, Millipore Sigma, Burlington, MA, USA) for 24 h. The human SIRT3 adenoviral vector (Ad-SIRT3) was obtained from Vector Biolabs (Malvern, PA, USA) and was infected into H9c2 cell lines. In brief, the H9c2 cells were incubated with Ad-SIRT3 at a dosage of 1 × 10^6^ PFU/mL (1:10) for 24 h before being added with Erastin (5 μM). Western blot analysis verified the increase in SIRT3 expression in H9c2 cells after Ad-SIRT3 incubation with/without Erastin (5 μM) (Appendix A).

### 2.3. Histological and Immunofluorescence Analysis

Serial sections were put in neutral-buffered 10% formalin solution (SF93–20; Fisher Scientific, Pittsburgh, PA, USA). Other sections were put in frozen OCT compound (4585; Fisher Health Care, Houston, TX, USA). In the same conditions, we cut these tissues to 10 µm in thickness. Paraffin parts were prepared for Masson’s trichrome staining and hematoxylin and eosin (H&E) staining. Some frozen sections were prepared for reactive oxygen species (ROS) measurement using DHE staining. Images were obtained using Nikon digital camera and analyzed in Nikon software (Nikon, Tokyo, Japan). Four to six random parts were picked for measurement using Image J (NIH, Bethesda, MD, USA).

### 2.4. Western Blot Analysis

After extraction, protein concentrations were tested using a BCA protein assay kit (Pierce Co, Rockford, IL, USA). Equal amounts of protein were run in 10% SDS-PAGE gel and transferred to a polyvinylidene difluoride (PVDF) membrane and then incubated with the primary antibodies at 4 °C overnight: β-myosin heavy chain (β-MHC; 1:1000, Abcam, Cambridge, MA, USA), α-smooth muscle actin (α-SMA; 1:1000, Abcam), 4-hydroxynonenal (4-HNE; 1:1000, Abcam), p53 acetylation (1:1000, Abcam), p53 (1:1000; Cell signaling, Danvers, MA, USA), glutathione peroxidase 4 (GPX-4) (1:1000; Novus Bio, Littleton, CO, USA) and TGF-β1 (1:500; Santa Cruz, CA, USA). After incubation with secondary antibody (1:5000; Santa Cruz), the signals were visualized and analyzed through image acquisition and analysis software (Bio-Rad, Hercules, CA, USA).

### 2.5. Iron-Level Measurement

The ferrous iron content in cells and tissues was measured using an iron assay kit (Abcam, Cambridge, MA, USA) based on manufacturer instructions [27].

### 2.6. Statistical Analysis

Data are presented as mean ± S.D. The significance of differences in the means of corresponding values among groups was determined by using the one-way ANOVA followed by Tukey post hoc tests. The significance of differences between two groups was determined using Student’s *t*-test. *p* < 0.05 was considered statistically significant. Data were analyzed using GraphPad Prism software, v.7.0 (GraphPad Software, La Jolla, CA, USA).

## 3. Results

### 3.1. Knockout of SIRT3 Led to Cardiac Remodeling and Ferroptosis in the Heart

Histological analysis and DHE staining showed that cardiac fibrosis, hypertrophy and ROS formation were significantly increased in the hearts of SIRT3KO mice (Figure 1A,B). This was accompanied by an increased expression of α-SMA (Figure 1C,D) and elevated 4-HNE level (Appendix A). Western blot analysis also showed that the expression of p53 and p53 acetylation was significantly increased, whereas the expression of GPX-4 was reduced in the hearts of SIRT3KO mice (Figure 1C,D).

### 3.2. SIRT3 Blunted Ferroptosis in Myofibroblasts via Suppression of p53 Acetylation

Erastin is widely used as an inducer of ferroptosis [28,29,30,31]. Overexpression of SIRT3 through Ad-SIRT3 treatment attenuated p53 acetylation (Figure 2A) and blunted Erastin-induced 4-HNE level in H9c2 myofibroblasts (Appendix A). Furthermore, overexpression of SIRT3 by Ad-SIRT3 blunted Erastin-induced reductions in GPX-4 expression in H9c2 cells (Figure 2A), suggesting SIRT3 as an important regulator of ferroptosis in myofibroblasts.

Next, we used C646 to pharmacologically block p53 acetylation [32,33] and to examine its role in ferroptosis. As shown in Figure 2B, treatment of H9c2 cells with C646 dramatically reduced the levels of acetylated p53. Moreover, C646 treatment significantly blunted Erastin-induced increases in the 4-HNE level (Appendix A) and ROS formation, as well as a reduction in GPX-4 expression (Figure 2B–D). Intriguingly, treatment of H9c2 cells with C646 did not significantly change ferrous iron levels among all groups (Figure 2D).

### 3.3. Mutations of p53 Acetylation Alleviated Ferroptosis in the Hearts of SIRT3KO Mice

P53^4KR^ mice were used to explore the role of acetylated p53 in ferroptosis [23]. In the heart tissue of SIRT3KO/p53^4KR^ mice, SIRT3 expression was absent (Appendix A), while p53 acetylation was decreased, followed by a significant reduction in interstitial fibrosis and cardiomyocyte size (Figure 3A,B). The levels of 4-HNE and ROS formation were reduced, whereas the expression of GPX-4 was upregulated in the hearts of SIRT3KO/p53^4KR^ mice compared to those of SIRT3KO mice (Figure 3C,D, Appendix A). There was an increase in ferrous iron levels, both in SIRT3KO and SIRT3KO/p53^4KR^ mice compared to that in WT mice. However, there was no difference in ferrous iron content between SIRT3KO mice and SIRT3KO/p53^4KR^ mice (Figure 3E).

### 3.4. Inhibition of Ferroptosis Reverses Cardiac Fibrosis in SIRT3-cKO Mice

Using SIRT3-cKO (cardiomyocyte-specific SIRT3 knockout) mice, we further examined whether inhibition of ferroptosis by ferrostatin-1 (Fer-1) attenuated cardiac fibrosis. Fer-1 treatment ameliorated cardiac ferroptosis, as indicated by reductions in the 4-HNE level (Appendix A) and ROS formation, as well as a downregulation of TGF-β1 expression in SIRT3-cKO mice (Figure 4A,B). Treatment of SIRT3-cKO mice with Fer-1 significantly reduced cardiomyocyte size and cardiac fibrosis, accompanied by decreases in expression of α-SMA and β-MHC (Figure 4C,D).

## 4. Discussion

In the present study, we aimed to explore the role of SIRT3-induced ferroptosis in myofibroblasts in cardiac fibrosis. Firstly, our data revealed a significant change in the expression of ferroptosis-associated genes and upregulation of p53 acetylation in the hearts of SIRT3KO mice. Secondly, overexpression of SIRT3 and inhibition of p53 acetylation reversed ferroptosis inducer-mediated ferroptosis in cultured H9c2 cells (myofibroblasts). Our data further demonstrate that SIRT3KO/p53^4KR^ mice had reduced levels of total p53 acetylation. Ferroptosis and cardiac fibrosis were blunted in SIRT3KO/p53^4KR^ mice. These results were further validated by the treatment of SIRT3-cKO mice with the ferroptosis inhibitor ferrostatin-1 (Fer-1). Treatment with Fer-1 ameliorated cardiac fibrosis, followed by reductions in lipid peroxidation, ROS and TGF-β1 levels. Our study strongly indicates that SIRT3/p53 acetylation-induced ferroptosis in myofibroblasts may contribute to cardiac fibrosis and remodeling (Figure 5).

Aging-associated diseases, such as hypertension, obesity and diabetes, bring a series of structural remodeling, including capillary rarefaction, cardiac fibrosis and hypertrophy [2]. Among these changes, cardiac fibrosis plays an essential role in the pathophysiology of diastolic dysfunction, ultimately leading to heart failure with preserved ejection fraction (HFpEF) [3]. Thus, the exploration of the underlying mechanisms behind how aging-associated diseases trigger cardiac fibrosis could lead to novel therapies in the clinic. SIRT3, residing in the mitochondria, is responsible for a series of cellular processes, such as energy homeostasis, oxidative stress and cell death [13]. Reduced SIRT3 levels have been observed in hypertensive, obese and diabetic patients, and are tightly associated with aging-associated diseases [17,34,35]. Our previous studies showed that absence of SIRT3 contributes to the development of cardiac fibrosis [4,14,17]. SIRT3 is defined as an imperative mediator between aging-associated diseases and cardiac fibrosis. Insight into the relationship between a lack of SIRT3 and cardiac fibrosis may provide new treatments for aging-induced cardiac dysfunction.

Previous studies already revealed that metabolic reprograming, inflammatory responses and an imbalance of oxidative stress induced by insufficiency or absence of SIRT3 result in the impairment or death of cardiomyocytes and endothelial cells as well as cardiac fibrosis. Using a specific pericyte-tracing animal model, we reported that the TGF-β1-ROS signaling pathway could mediate pericyte–myofibroblast transition, which plays an important role in cardiac fibrosis in SIRT3KO mice [17]. While the effects of non-fibrotic cells on fibrosis were well documented [36,37,38], few studies were focused on the fibrotic cells themselves [6]. A recent study proposed the death of myofibroblasts as a crucial driver of cardiac fibrosis [7]. Apart from other forms of cell death, ferroptosis is a newly found one mainly characterized by ferrous iron accumulation and lipid peroxidation [10]. Lipid peroxides such as 4-HNE released from myofibroblasts stimulate the secretion of TGF-β1 from surrounding cells, which induces cardiac fibrosis and, subsequently, diastolic dysfunction [11,12]. SIRT3 involved in energy metabolism and oxidative stress in the mitochondria may play a crucial role in ferroptosis [15,17,39]. Our present study, for the first time, provides evidence, which indicates a possible correlation between SIRT3 and ferroptosis. Elevation of 4-HNE levels and reduced GPX-4 levels were found in SIRT3KO mice compared to WT mice, suggesting ferroptosis as an imperative character in cardiac fibrosis upon absence of SIRT3 (Figure 1C,D, Appendix A). H9c2 cells were used as cardiomyocytes in most studies previously, while more and more researchers propose it as an in vitro cell model for myofibroblasts [40]. Our data showed that overexpression of SIRT3 in H9c2 cells attenuates Erastin-induced increases in 4-HNE and a reduction in GPX-4 levels (Figure 2A, Appendix A), further supporting a regulatory role of SIRT3 in the ferroptosis of myofibroblasts. Taken together, our study suggests that ferroptosis in myofibroblasts in the absence of SIRT3 may function as a major contributor to cardiac fibrosis and diastolic dysfunction. So far, the underlying mechanisms of SIRT3 deficiency-induced ferroptosis remain elusive.

Our present data showed that levels of p53 and p53 acetylation were elevated in the hearts of SIRT3KO mice compared to those of WT mice (Figure 1C,D). P53 is well known for its beneficial role as a tumor suppressor [22]. On the other hand, the functional roles of p53 in injured hearts were mainly detrimental via promoting cell death, such as apoptosis, autophagy and necrosis [13,41,42]. The suppression of p53 in cardiomyocytes and endothelial cells was considered to be an efficient method to constrain the development of cardiac fibrosis [43,44]. In contrast, the contributions of p53 in myofibroblasts are controversial. A previous study revealed that increased p53-induced apoptosis blunts the proliferation of fibroblasts, thus leading to a reduction in fibrosis [45]. Meanwhile, other studies demonstrated p53 to be an activator of fibroblast proliferation via the SMAD3 pathway, promoting cardiac fibrosis [33,46]. These studies implicate the essential roles of p53 in fibrotic cells in the pathogenesis of cardiac fibrosis. In general, p53 is mainly activated by phosphorylation, ubiquitination and acetylation [41,42]. Gu et al. identified that acetylated p53 works as a crucial regulator for ferroptosis in tumor development, and acetylated p53/GPX-4 was proposed to be a canonical ferroptosis signaling pathway [47,48]. Consistent with this study, our results showed that overexpression of SIRT3 attenuates acetylation of p53 and increases GPX-4 expression in H9c2 cells (Figure 2A), while knockout of SIRT3 increases p53 acetylation and ferroptosis in mouse hearts (Figure 3B,D, Appendix A), indicating SIRT3/acetylated p53/GPX-4 as a crucial ferroptosis pathway in myofibroblasts. C646 was defined as a suppressor of p53 acetylation in tumors [32,33]. As we expected, Erastin-induced ferroptosis was suppressed by treatment with C646, as evidenced by decreased levels of 4-HNE and ROS formation and increased levels of GPX-4 (Figure 2B–D, Appendix A). p53^4KR^ mice have been used as an animal model for tumors, which are incapable of activating the ferroptosis pathway due to mutation of specific acetylation sites of p53 [23,48]. To explore the role of acetylated p53 and absence of ferroptosis in SIRT3KO-mediated cardiac fibrosis, we generated a novel SIRT3KO/p53^4KR^ mouse model. Our results show that total p53 acetylation was reduced in SIRT3KO/p53^4KR^ mice. Furthermore, ferroptosis marker 4-HNE was reduced while the expression of GPX-4 was upregulated in SIRT3KO/p53^4KR^ mice compared to that of SIRT3KO mice. This was accompanied by a suppression of cardiac fibrosis and cardiac hypertrophy (Figure 3A). Data from our present study strongly suggest that SIRT3-acetylated p53 signaling, which induced ferroptosis in myofibroblasts, may contribute to cardiac fibrosis and hypertrophy in SIRT3KO mice.

In addition, there is a significant increase in α-SMA expression in the cardiomyocyte-specific SIRT3 knockout (SIRT3cKO) mice compared to WT mice. TGF-β1 was also upregulated in the heart of SIRT3cKO mice as compared to that of WT mice (Figure 4B). The expression of α-SMA and TGF-β1 was widely used as biomarkers for myofibroblasts [49], indicating increased numbers of myofibroblasts induced by cardiomyocyte loss of SIRT3. So far, little is known about the proliferation of myofibroblasts in cardiac fibrosis. Our data revealed that ferroptosis of myofibroblasts was also initiated in the absence of cardiomyocyte SIRT3. In addition to endothelial cells and pericytes, TGF-β1, the key factor for proliferation/transition of myofibroblasts, was reported to be synthesized in cardiomyocytes [50,51]. Consistent with these findings, the levels of TGF-β1 were significantly elevated in SIRT3-cKO mice. Kim et al. revealed that TGF-β1 could be induced via lipid peroxidation, which is the core of ferroptosis [52]. This notion was supported by our findings that treatment with ferrostatin-1 (Fer-1) attenuated levels of TGF-β1 as well as lipid peroxidation and ferroptosis in SIRT3-cKO mice (Figure 4B,C, Appendix A). The alleviation by Fer-1 indicated that absence of SIRT3 in cardiomyocytes triggers TGF-β1 through ferroptosis, which further induced proliferation or transition of myofibroblasts and contributed to cardiac fibrosis.

Overall, our findings expand the importance of the SIRT3/acetylated p53 pathway-induced ferroptosis in myofibroblasts as a potential therapeutic target for inhibiting further development of cardiac fibrosis, which may contribute to cardiac dysfunction in aging individuals.

## Figures and Tables

**Figure 1 cells-12-01428-f001:**
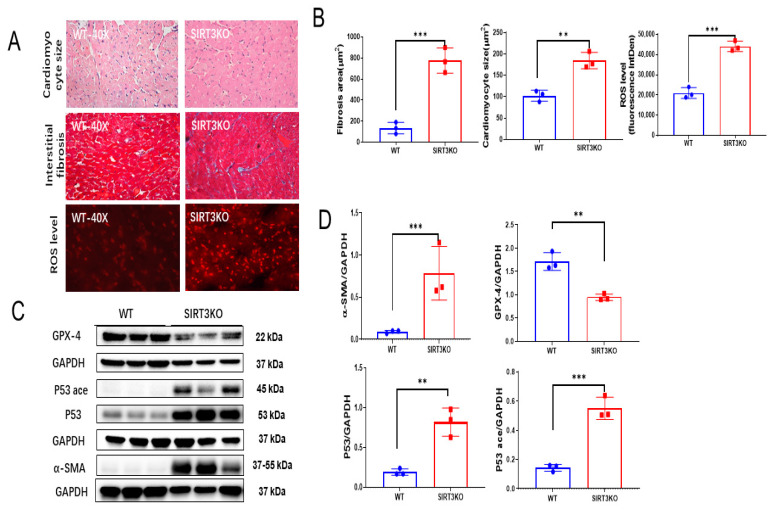
Loss of SIRT3 resulted in cardiac remodeling and ferroptosis. (**A**) Representative images of H&E, Masson’s trichrome and DHE stains of whole heart sections in WT mice and SIRT3KO mice. (**B**) Quantification of cardiomyocyte sizes, interstitial fibrosis area and fluorescence-integrated density of DHE assay in the indicated groups (*n* = 3). (**C**,**D**) Immunoblots and analysis of α-SMA, p53, acetylated p53, GPX-4 and corresponding GAPDH in the indicated mouse hearts (*n* = 3–4). Mean ± S.D., ** *p* < 0.01, *** *p* < 0.001.

**Figure 2 cells-12-01428-f002:**
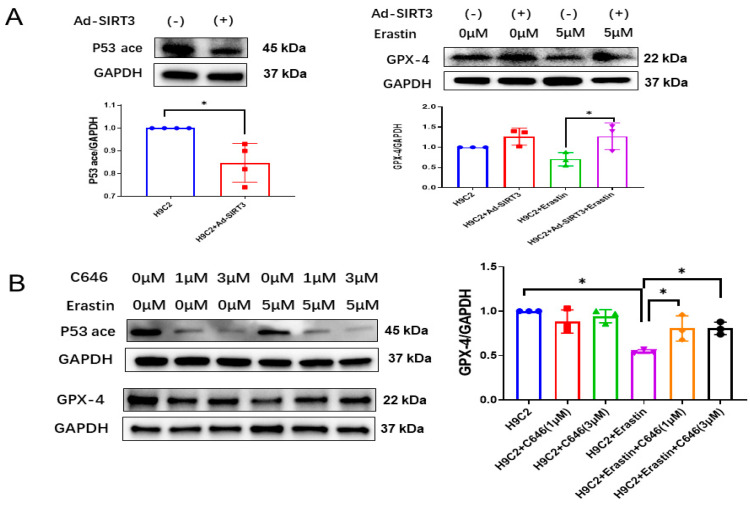
SIRT3-acetylated p53 mediates ferroptosis in H9c2 myofibroblasts. (**A**) Immunoblots and analysis of p53 acetylation, GPX-4 and GAPDH in H9c2 cells treated with Ad-SIRT3 alone or treated with Ad-SIRT3 and Erastin (*n* = 3). (**B**) Immunoblots and analysis of p53 acetylation, GPX-4 and GAPDH in H9c2 cells treated with/without Erastin and C646 (*n* = 3). (**C**) Representative images of DHE-stained H9c2 cells treated with/without Erastin and C646. (**D**) Quantification of ROS fluorescence integrated density and ferrous OD value in the indicated H9c2 cells treated with/without Erastin and C646 (*n* = 3). Mean ± S.D., * *p* < 0.05, ** *p* < 0.01.

**Figure 3 cells-12-01428-f003:**
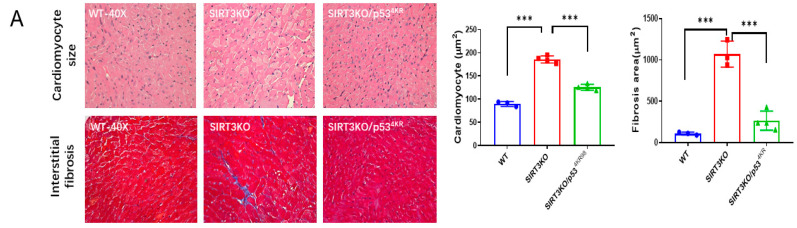
Inhibition of acetylated p53 rescued ferroptosis and cardiac fibrosis in SIRT3KO mice. (**A**) Representative images of H&E-stained and Masson’s trichrome-stained whole heart sections and quantification of cardiomyocyte sizes and interstitial fibrosis area in WT mice, SIRT3KO mice and SIRT3KO/p534KR mice (*n* = 3–4). (**B**) Immunoblots and analysis of α-SMA, p53, p53 acetylation and GAPDH in the indicated mouse hearts (*n* = 3–4). (**C**) Representative images of DHE-stained whole heart sections in the indicated mouse hearts. (**D**) Immunoblots and analysis of GPX-4 and GAPDH ratio in the indicated mouse hearts (*n* = 3–4). (**E**) Quantification of ferrous iron OD value in the indicated H9c2 cells (*n* = 3). Mean ± S.D., * *p* < 0.05, ** *p* < 0.01, *** *p* < 0.001.

**Figure 4 cells-12-01428-f004:**
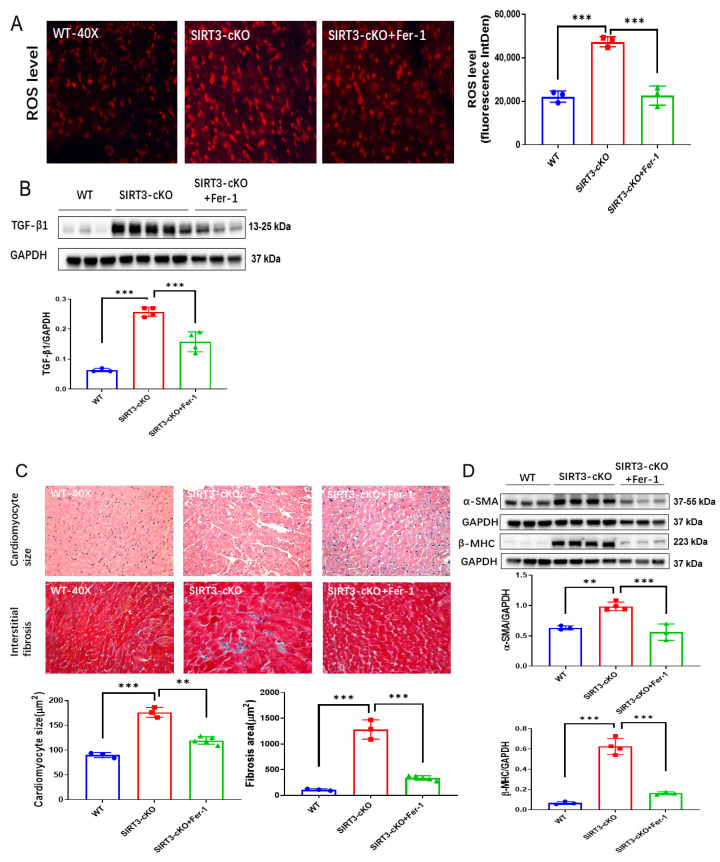
Inhibition of ferroptosis by Fer-1 reversed cardiac fibrosis in SIRT3cKO mice. (**A**) Representative images of DHE-stained whole heart sections and quantification of fluorescence-integrated density in WT mice, SIRT3-cKO mice and SIRT3-cKO+Fer-1 mice (*n* = 3). (**B**) Immunoblots and analysis of TGF-β1 and GAPDH in the indicated mouse hearts (*n* = 3–4). (**C**) Images of H&E- and Masson’s trichrome-stained whole heart sections and quantification of cardiomyocyte sizes and interstitial fibrosis area in the indicated groups (*n* = 3–5). (**D**) Immunoblots and analysis of α-SMA, β-MHC and GAPDH in the indicated mouse hearts (*n* = 3–4). Mean ± S.D., ** *p* < 0.01, *** *p* < 0.001.

**Figure 5 cells-12-01428-f005:**
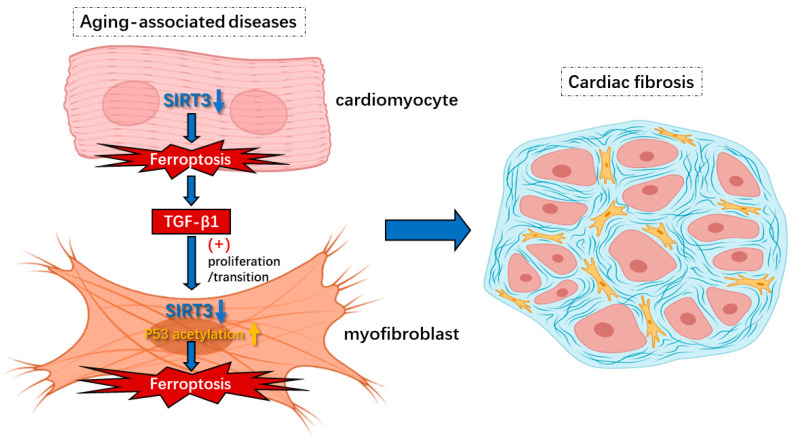
Knockout of SIRT3-induced ferroptosis in cardiomyocytes and myofibroblasts under aging-associated diseases contributes to cardiac fibrosis. Insufficiency of SIRT3 in aging-associated diseases could trigger ferroptosis via hyperacetylation of p53 in myofibroblasts, which leads to increased release of lipid peroxides such as 4-HNE that cause secondary impairments. In addition, the induction of ferroptosis in cardiomyocytes from lack of SIRT3 results in enhanced levels of TGF-β1 that contribute to proliferation/transition of myofibroblasts. Overall, ferroptosis in cardiomyocytes and myofibroblasts caused by reduction in SIRT3 results in cardiac fibrosis and diastolic dysfunction.

## Data Availability

The authors declare that all supporting data are available within the article.

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
