# Peer review of "SIRT3 Deficiency Enhances Ferroptosis and Promotes Cardiac Fibrosis via p53 Acetylation"

_cells, 2023, doi:10.3390/cells12101428_

Round 1

Reviewer 1 Report

The manuscript by Su et al. aims to decipher the role of p53 acetylation in ferroptosis and cardiac fibrosis induced by SIRT3-deficiency.

For this, the authors analyse the regulation of several ferroptosis markers, as well as acetylated p53, after SIRT3 KO or by pharmacological induction of ferroptosis. The role of p53 acetylation in this context is analyzed by pharmacological inhibition and the generation of SIRT3KO/p534KR compound mutant mice. Finally, the role of SIRT3 deficiency-induced ferroptosis in cardiac fibrosis is analyzed using conditional SIRT3 KO mice and concomitant inhibition of ferroptosis by ferrostatin-1.

The SIRT3-acetylated p53-ferroptosis pathway is an interesting topic. However, many of the findings presented here have in principle been published previously, resulting in only limited novelty of the manuscript.

Major points :

Fig. 1-4 : The organization of figures on two consecutive pages, as provided in the manuscript .pdf file,  is not acceptable. The labelling of subfigures C and D is missing, making the reading of the figures confusing.

Fig.2A (and B) do not show Erastin-induced elevation of acetylated p53, as claimed in the text (line 132), but rather a reduction.

The function of the proteins used for western blot analyses (e.g. GPX-4) has to be explained earlier in the manuscript.

The generation of Ad-SIRT3 has to be described in the materials and methods section.

Minor points :

Line 55 : Please change to „its acetylation“ or „p53 acetylation“

Line 61 : Please re-phrase „SIRT3-induced p53 acetylation“. SIRT3 is a deacetylase.

Author Response

Major points :

Fig. 1-4 : The organization of figures on two consecutive pages, as provided in the manuscript .pdf file,  is not acceptable. The labelling of subfigures C and D is missing, making the reading of the figures confusing.  

We thank the reviewer for these constructive comments. These issues have been resolved. All the figures are replaced with new figures.

Fig.2A (and B) do not show Erastin-induced elevation of acetylated p53, as claimed in the text (line 132), but rather a reduction.  

We agree with the reviewer’s comments that erastin did not significantly altered p53 acetylation. We have deleted the erastin-induced p53 acetylation, but kept Ad-Sirt3 on p53 acetylation in Fig.2 A.

The function of the proteins used for western blot analyses (e.g. GPX-4) has to be explained earlier in the manuscript.

Done as suggested. We used GPX-4 as ferroptosis marker.

The generation of Ad-SIRT3 has to be described in the materials and methods section.

We obtained Ad-SIRT3 from vector Biolabs (Malvern, PA 19355). This information was included in the methods.

Minor points :

Line 55 : Please change to „its acetylation“ or „p53 acetylation  “

We have corrected it

Line 61 : Please re-phrase „SIRT3-induced p53 acetylation“. SIRT3 is a deacetylase. 

We have  corrected it

Reviewer 2 Report

In their manuscript “SIRT3 deficiency enhances ferroptosis and promotes cardiac fibrosis via p53 acetylation”, Han Su et al. investigate the role of SIRT3 in cardiac ferroptosis and p53 acetylation. They found that SIRT3KO mice have an increased cardiac fibrosis and ferroptosis. Moreover, they propose that SIRT3-mediated cardiac fibrosis is caused by SIRT3-mediated p53 acetylation in myofibroblasts.

The manuscript is well written, and the data are interesting. However, the role of SIRT3 for p53-mediated ferroptosis has been already demonstrated (PMID: 33377976, not cited by the authors) and the authors do not provide any additional mechanism to explain their findings. Additional experiments are required to clarify the role of p53 acetylation in the authors findings. Moreover, some important controls are missing to support the experiments.

Major comments:

1. Figure 2 does not demonstrate that SIRT3-acetylated p53 mediates ferroptosis in myofibroblasts as proposed by the authors. C646 is a p300 inhibitor and not a specific inhibitor of p53 acetylation. Experiments should be performed in absence of p53 (for instance, by silencing it in H9c2 cells) to validate its implication.

2. In the same line, analysis of the expression of ferroptosis-related p53 targets such as SLC7A11 upon KO or overexpression of SIRT3 in myofibroblasts would be needed to support the authors conclusions on the role of p53 acetylation in the SIRT3-mediated regulation of ferroptosis.

3. The authors base their work on SIRT3KO animal models and SIRT3 overexpression mediated by adenoviruses. However, there is not a single control of SIRT3 expression in all the experiments. Western blots showing SIRT3 protein level or its absence should be shown in figures 1B, 2A, 3A, 4A.

4. The authors cannot show only a single selected protein bands for 4HNE western blots and conclude about lipid peroxidation. The full length of the western blot should be shown and quantified. Looking at the whole protein peroxidation profile in supplementary data, the results are a lot less striking than what the authors suggest.

Minor comments:

5. Methods should be better described. For instance, SIRT3 adenoviruses are not sufficiently detailed.

6. The authors mention several times that Erastin induces the “expression of 4-HNE” (line 131, line 137). 4-HNE is a product of lipid peroxidation which forms adducts on proteins, not a gene that is expressed. The authors should reformulate.

Author Response

Major comments:

  1. Figure 2 does not demonstrate that SIRT3-acetylated p53 mediates ferroptosis in myofibroblasts as proposed by the authors. C646 is a p300 inhibitor and not a specific inhibitor of p53 acetylation. Experiments should be performed in absence of p53 (for instance, by silencing it in H9c2 cells) to validate its implication.

We agree with the review’s comments that C646 is not specific inhibitor for p53 acetylation. P53 acetylation is p300 dependent [1]. C646, as a p300 inhibitor, is surely not specific for p53 acetylation, while under limited circumstances that we can’t specific knockdown of p53 acetylation in H9c2 cells. Knockdown of p53 may not abolish p53 acetylation, therefore, C646 could be the most practical way to suppress p53 acetylation. In addition, several papers used C646 as the inhibitor of p53 acetylation [2,3]. Moreover, the results in vivo were consistent with the results in vitro, which, to some extent, verified the function of p53 acetylation in ferroptosis of myofibroblasts.

  1. In the same line, analysis of the expression of ferroptosis-related p53 targets such as SLC7A11 upon KO or overexpression of SIRT3 in myofibroblasts would be needed to support the authors conclusions on the role of p53 acetylation in the SIRT3-mediated regulation of ferroptosis.

We agree with reviewer’s comments. SLC7A11 is definitively a classic mediator between p53 and ferroptosis, meanwhile it is not the only one. SAT1 (spermidine/spermine N1-acetyltransferase 1) and GLS2 (glutaminase 2) are also crucial mediator between p53 and ferroptosis [4]. SLC7A11 might be just part of p53 acetylation-ferroptosis pathway. And we have found the alterations in GPX-4, which illustrate the impaired capacity of scavenging lipid peroxidation could be the mechanism of how p53 acetylation influence ferroptosis. Furthermore, though we didn’t investigate the specific mechanisms of p53 acetylation-ferroptosis pathway, the p53 acetylation/GPX-4/ferroptosis signaling in myofibroblast was firstly proposed by us, which made this paper interesting and inspiring.  

  1. The authors base their work on SIRT3KO animal models and SIRT3 overexpression mediated by adenoviruses. However, there is not a single control of SIRT3 expression in all the experiments. Western blots showing SIRT3 protein level or its absence should be shown in figures 1B, 2A, 3A, 4A.

We thank the reviewer for these comments. We have added these data in supplementary figure 2

  1. The authors cannot show only a single selected protein bands for 4HNE western blots and conclude about lipid peroxidation. The full length of the western blot should be shown and quantified. Looking at the whole protein peroxidation profile in supplementary data, the results are a lot less striking than what the authors suggest.

We agree with reviewer’s comments. We have reexamined these data. Please see supplemental Figure 1.

Minor comments:

  1. Methods should be better described. For instance, SIRT3 adenoviruses are not sufficiently detailed.

The detail information was added in materials and methods section.

  1. The authors mention several times that Erastin induces the “expression of 4-HNE” (line 131, line 137). 4-HNE is a product of lipid peroxidation which forms adducts on proteins, not a gene that is expressed. The authors should reformulate.

Thank the reviewer for these comments. We have corrected this in the reversion.

[1]        Sebti S, Prébois C, Pérez-Gracia E, et al. BAT3 modulates p300-dependent acetylation of p53 and autophagy-related protein 7 (ATG7) during autophagy. Proc Natl Acad Sci U S A, 2014, 111(11): 4115-4120.

[2]        Zheng S, Koh XY, Goh HC, et al. Inhibiting p53 Acetylation Reduces Cancer Chemotoxicity. Cancer Res, 2017, 77(16): 4342-4354.

[3]        Gao XY, Lai YY, Luo XS, et al. Acetyltransferase p300 regulates atrial fibroblast senescence and age-related atrial fibrosis through p53/Smad3 axis. Aging Cell, 2023, 22(1): e13743.

[4]        Kang R, Kroemer G, Tang D. The tumor suppressor protein p53 and the ferroptosis network. Free Radic Biol Med, 2019, 133: 162-168.

Round 2

Reviewer 1 Report

Most points raised by this reviewer have been addressed by the authors.

Minor points:

Line 55 : please change to "and its acetylation”. Otherwise it is not clear to what "acetylation" actually refers to.

Line 150 : please correct GXP-4

Reviewer 2 Report

The authors have corrected their analysis of 4-HNE staining and added the necessary controls for SIRT3 levels. Even if the authors did not deem necessary to perform the suggested additional experiments on p53 and its targets, overall, the study is robust enough to be acceptable for publication.